# Enhancing Generalization via Sharpness-Aware Trajectory Matching for Dataset Condensation

## Abstract

Dataset condensation aims to synthesize datasets with a few representative samples that can effectively represent the original datasets. This enables efficient training and produces models with performance close to those trained on the original sets. Most existing dataset condensation methods conduct dataset learning under the bilevel (inner- and outer-loop) based optimization. However, the preceding methods perform with limited dataset generalization due to the notoriously complicated loss landscape and expensive time-space complexity of the inner-loop unrolling of bilevel optimization. These issues deteriorate when the datasets are learned via matching the trajectories of networks trained on the real and synthetic datasets with a long horizon inner-loop. To address these issues, we introduce Sharpness-Aware Trajectory Matching (SATM), which enhances the generalization capability of learned synthetic datasets by optimizing the sharpness of the loss landscape and objective simultaneously. Moreover, our approach is coupled with an efficient hypergradient approximation that is mathematically well-supported and straightforward to implement, along with controllable computational overhead. Empirical evaluations of SATM demonstrate its effectiveness across various applications, including in-domain benchmarks and out-of-domain settings. Moreover, its easy-to-implement properties afford flexibility, allowing it to integrate with other advanced sharpness-aware minimizers.

## 1 Introduction

The success of modern deep learning in various fields, exemplified by Segment Anything (Kirillov et al., 2023) in computer vision and GPT (Ouyang et al., 2022) in natural language processing, comes at a significant cost in terms of the enormous computational expenses associated with large-scale neural network training on massive amounts of real-world data Radford et al. (2021); Li et al. (2023); Schuhmann et al. (2022); Li et al. (2022); Gowda et al. (2023). To reduce training and dataset storage costs, selecting the representative subset based on the specific importance criteria forms a direct solution (Har-Peled & Mazumdar, 2004; Yang et al., 2022; Paul et al., 2021; Wang et al., 2022b). However, these methods fail to handle the cases when the samples are distinct and the information is uniformly distributed in the dataset. In contrast, Dataset Condensation (DC) (Zhao et al., 2021; Zhao & Bilen, 2023; Wang et al., 2018; Cazenavette et al., 2022; Du et al., 2023) focuses on creating a small, compact version of the original dataset that retains its representative qualities. As a result, models trained on the condensed dataset perform comparably to those trained on the full dataset, significantly reducing training costs and storage requirements, and meanwhile expediting other machine learning tasks such as hyperparameter tuning, continual learning (Rosasco et al., 2021), architecture search (Sangermano et al., 2022; Yu et al., 2020; Masarczyk & Tautkute, 2020), and privacy-preserving (Shokri & Shmatikov, 2015; Dong et al., 2022).

Given the significant practical value of condensed datasets, considerable effort has been directed toward designing innovative surrogate methods to ensure that synthetic datasets capture representative characteristics, thereby enhancing future deployments' performance (Zhao & Bilen, 2023; Zhao et al., 2021; Zhou et al., 2022; Kim et al., 2022). Bilevel optimization (BO) provides a DC paradigm for learning synthetic datasets through its main optimization objective in the outer-loop constrained by training neural networks in its inner-loop. One line of the representative solutions condenses

datasets by minimizing the disparity between training trajectories on synthetic and real sets, achieving notable performance (Cazenavette et al., 2022). The following studies either reduce the computational cost of inner-loop unrolling or steer the optimization process to enhance the generalization of the learned dataset to the unseen tasks. For instance, FTD (Du et al., 2023) improves the performance of synthetic datasets by leveraging high-quality inner-loop expert trajectories and incorporating momentum into the outer-loop optimization via Exponential Moving Average (EMA) with extra memory overhead introduced, increasing along with the synthetic dataset budget. TESLA (Cui et al., 2023) is proposed with a two-inner-loop-based algorithm to approximate the hypergradient for the dataset updates maintaining a constant memory usage. However, the outer-loop loss landscape, formed with numerous sharp regions shaped by the dynamics of the inner-loop (Abbas et al., 2022; Franceschi et al., 2017), is often overlooked in the dataset condensation field. The challenges inherent in such optimization settings result in the limited generalization performance of the learned datasets.

Inspired by the sharpness-aware optimizers (Foret et al., 2020; Kwon et al., 2021; Li & Giannakis, 2024), which improve generalization by minimizing loss landscape sharpness to achieve flat convergence regions in uni-level optimization, we propose Sharpness-Aware Trajectory Matching (SATM) to DC to reduce the sharpness of the outer-loop landscape and enhance the generalization ability of the learned dataset. However, direct application is infeasible due to the tremendous computation overhead caused by the notorious two-stage gradient estimation, which typically doubles both the time and memory costs throughout the learning process. To address this issue, we propose a lightweight trajectory matching-based method composed of two computationally efficient strategies, namely truncated unrolling hypergradient and trajectory reusing, with controllable memory cost for gradient estimation. Our method improves the generalization ability of the trajectory-matching algorithm significantly on both in-domain and out-of-domain tasks with noticeable improvement margins across various applications, whilst achieving efficient time and memory cost. More specifically, in terms of the computational overhead, our method surpasses one of the most efficient algorithms, TESLA (Cui et al., 2023), regarding runtime cost with comparable memory consumption. The main contributions of this work are summarised as:

- We primarily study and improve the generalization ability of dataset condensation and propose Sharpness-Aware Trajectory Matching by jointly minimizing the sharpness and the distance between training trajectories with a tailored loss landscape smoothing strategy.

- A simple and easy-to-implement method, integrating two hypergradient approximation strategies, is proposed to handle the tremendous computational overhead introduced by sharpness minimization. We further reduce the computational redundancy by deriving a closed-form solution for the learning rate learning. For all the proposed approximation methods, we provide rigorous theoretical support by bounding the errors of the approximations and analysing the approximation error caused by hyperparameters, which shed light on meaningful hyperparameter tuning.

- SATM outperforms the trajectory-matching-based competitors on various dataset condensation benchmarks under in- and out-of-domain settings. More importantly, our method demonstrates noticeable improvement margins on ImageNet-1K where most existing methods fail to condense.

## 2 RELATED WORK

### 2.1 DATASET CONDENSATION

Inspired by knowledge distillation (Gou et al., 2021; Yang et al., 2020) and meta-learning driven by Bilevel optimization (BO) (Lorraine et al., 2020; Maclaurin et al., 2015; MacKay et al., 2019; Finn et al., 2017; Gao et al., 2022; Rajeswaran et al., 2019; Gao et al., 2021), Wang *et al.* (Wang et al., 2018) leverage BO to distill a small, compact synthetic dataset for efficient training on unseen downstream tasks. Several works expanding on this framework match gradients (Zhao & Bilen, 2021; Zhao et al., 2021; Lee et al., 2022), features (Wang et al., 2022a), and distributions (Zhao & Bilen, 2023) produced by the synthetic and real sets. RDED (Sun et al., 2024) introduces new perspectives to the dataset distillation field by constructing synthetic images from original image crops and labelling them with a pre-trained model. Usually, the existing dataset condensation methods conduct a few iterations of inner-loop unrolling in BO to mitigate the computational cost of the nested optimization

process. To avoid the same issue, Nguyen *et al.* (Nguyen et al., 2021b;a) directly estimate the convergence of the inner-loop using the Neural Tangent Kernel (NTK) to emulate the effects from the synthetic sets. However, due to the heavy computational demands of matrix inversion, the NTK-based method struggles to scale up for condensing large, complex datasets. MTT (Cazenavette et al., 2022) emphasises the benefits of a long horizon inner-loop and minimizes the differences between synthetic and expert training trajectory segments with the following studies such as FTD (Du et al., 2023), TESLA (Cui et al., 2023), and DATM (Guo et al., 2024). Nonetheless, the learned synthetic dataset often overfits the neural architecture used in the expert trajectories, resulting in limited generalization ability. In this work, we address this problem by exploring the flatness of the synthetic dataset's loss landscape.

## 2.2 FLATNESS OF THE LOSS LANDSCAPE AND GENERALIZATION

The generalization enhanced by flat region minimums has been observed empirically and studied theoretically (Dinh et al., 2017; Keskar et al., 2016; Neyshabur et al., 2017; Tahmasebi et al., 2024). Motivated by this, Sharpness-aware minimizer (SAM) (Foret et al., 2020) optimizes the objective function and sharpness simultaneously to seek the optimum lying in a flat convergence region. However, the computational overhead of SAM is double that of the conventional optimization strategy. To address this issue, ESAM (Du et al., 2022) randomly selects a subset of the parameters to update in each iteration. Zhuang *et al.* (Zhuang et al., 2021) observes that SAM fails to identify the sharpness and mitigates this by proposing a novel sharpness proxy. To tackle the complicated loss landscape, Li and Giannakis (Li & Giannakis, 2024) introduce a momentum-like strategy for sharpness approximation while ASAM (Kwon et al., 2021) automatically modify the sharpness reaching range by adapting the local loss landscape geometry. In contrast, we handle complicated multi-iteration unrolling for learning datasets in the many-shot region with both the difficulty of the sharpness approximation and the surge in computation resources.

## 3 PRELIMINARY

### 3.1 DATASET CONDENSATION AND MATCHING TRAINING TRAJECTORY

Dataset condensation focuses on synthesizing small datasets with a few representative samples that effectively capture the essence of the original datasets. Cazenavette et al. (2022) proposed to create the synthetic datasets by minimizing the distance between the training trajectory produced by the synthetic set, named synthetic trajectories, and those by the real set, termed expert trajectories, with the assumption that the datasets containing similar information generate close training trajectories, *a.k.a*, matching training trajectory (MTT). A sequence of expert weight checkpoints, $\theta_t^E$, are collected during the training on the real sets in the order of iterations, $t$, to construct the expert trajectories, $\{\theta_t^E\}_{t=0}^T$ with $T$ denoting the total length of the trajectory. The pipeline of MTT begins with sampling a segment of expert trajectory, starting from $\theta_t^E$ to $\theta_{t+M}^E$ with $0 \leq t \leq t + M \leq T$. Then, to generate a synthetic segment, a model, $\theta_t^S$, is initialised by, $\theta_t^E$, and trained on the learnable dataset, $\phi$, to get $\theta_{t+N}^S(\phi)$ after $N$ iteration. Following the bilevel optimization context, the disparity between $\theta_{t+N}^S(\phi)$ and $\theta_{t+M}^E$ is optimized to learn datasets with the outer-loop objective as:

$$\min_\phi \mathcal{L}(\theta^S(\phi)) := \frac{1}{\delta}||\theta_{t+N}^S(\phi) - \theta_{t+M}^E||_2^2 \quad \text{s.t.} \ \theta_{t+N}^S(\phi) = \Xi_N(\theta_t^S, \phi)$$

where $\Xi_N(\cdot)$ represents $N$ differentiable minimizing steps on the inner-loop objective, CrossEntropy loss, $\mathcal{L}_{CE}(\theta, \phi)$. The existing optimizers can instantiate the inner-loop, such as SGD whose one-step optimization is exemplified by $\Xi(\theta, \phi) = \theta - \alpha \nabla \mathcal{L}_{CE}(\theta, \phi)$ where $\alpha$ denotes the learning rate. Note that $M$ and $N$ are not necessarily equal since dense information in the synthetic datasets leads to fast training. $\delta$ in Eq. 1, stabilising the numerical computation, can be unpacked as $||\theta_t^E - \theta_{t+M}^E||_2^2$.

### 3.2 SHARPNESS-AWARE MINIMIZATION

Given the training data, $D$, consider a training problem where the objective function is denoted as $\mathcal{L}(\phi; D)$ with the learnable parameter $\phi$, the objective function of SAM is framed as:

$$\min_\phi \max_{||\epsilon||_2 \leq \rho} \mathcal{L}(\phi + \epsilon; D), \tag{1}$$

---

**Algorithm 1** Sharpness-Aware Trajectory Matching for dataset condensation.

---

1: **Input:** $\{\theta_t^E\}_0^T$, $\alpha$, $\beta$.
2: **Output:** $\phi$
3: Init $\phi$
4: **while** not converged or reached max steps **do**
5:     Sample an iteration $t$ to construct an expert segment, $\theta_t^E$, and $\theta_{t+M}^E$
6:     $\theta^S = \theta_t^E$
7:     $\phi_j^\Delta \sim \mathcal{N}(0, \gamma||\phi_j||_2 I)$
8:     $\phi = \phi + \phi^\Delta$
9:     **for all** $i \leftarrow 1$ to $N$ **do**
10:         $\theta^S = \theta^S - \alpha\nabla\mathcal{L}(\theta^S, \phi)$
11:     **end for**
12:     Compute $\nabla F(\phi)$ by Eq. 6
13:     $\epsilon = \rho\nabla F(\phi)/||\nabla F(\phi)||_2$
14:     $\bar{\theta}^S = \theta_{t+\kappa}^S$
15:     $\phi = \phi - \phi_\Delta$
16:     **for all** $i \leftarrow N - \tau$ to $N$ **do**
17:         $\bar{\theta}^S = \bar{\theta}^S - \alpha\nabla\mathcal{L}(\bar{\theta}^S, \phi + \epsilon)$
18:     **end for**
19:     Compute $\nabla F(\phi + \epsilon)$ by Eq. 7
20:     $\phi = \phi - \beta\nabla F(\phi + \epsilon)$
21: **end while**

---

where approximating sharpness is achieved by finding the perturbation vectors $\epsilon$ maximizing the objective function in the Euclidean ball with radius, $\rho$, with the sharpness defined as:

$$\max_{||\epsilon||_2 \leq \rho} \left| \mathcal{L}(\phi + \epsilon; D) - \mathcal{L}(\phi; D) \right|. \tag{2}$$

Instead of solving this problem iteratively, a closed-form approximation of the optimality by utilisation of the first-order Taylor expansion of the training loss is given by

$$\epsilon = \rho\frac{\nabla\mathcal{L}(\phi)}{||\nabla\mathcal{L}(\phi)||_p} \approx \arg\max_{||\epsilon|| \leq \rho} \mathcal{L}(\phi + \epsilon).$$

Overall, the updating procedure of SAM in each iteration is summarised as follows:

$$\phi = \phi - \alpha\nabla\mathcal{L}(\phi + \epsilon) \quad \textbf{s.t.} \quad \epsilon = \rho\frac{\nabla\mathcal{L}(\phi)}{||\nabla\mathcal{L}(\phi)||_p}, \tag{3}$$

where $\alpha$ represents the learning rate and after computing the gradient, $\nabla\mathcal{L}(\phi + \epsilon)$, the parameter update procedure is instantiated by standard optimizers, such as SGD and Adam (Kingma & Ba, 2015). Without losing generality, we set $p = 2$ for simplicity for the rest of this work. Due to the two-stage gradient calculation at $\phi$ and $\phi + \epsilon$, the computational overhead of SAM is doubled.

## 4 METHOD

We introduce our method in this section by starting with configuring the trajectory matching-based dataset condensation under the Sharpness-Aware Bilevel optimization framework while handling the inaccurate sharpness approximation. Then, two strategies with mathematical support are proposed to reduce the computation cost introduced by the vanilla application of SAM. Additionally, we further boost the computational efficiency with a closed-form solution for learning rate learning instead of the backpropagation through inner-loop unrolling. The general idea is summarised in Algorithm 1.

### 4.1 SMOOTH SHARPNESS-AWARE MINIMIZATION FOR DATASET CONDENSATION

Generalizing to the unseen tasks is challenging for the learned synthetic datasets. To mitigate this issue, we steer the optimization on the outer-loop in Eq. 1 and minimize the objective function forward landing in the flat loss landscape region to enable the synthetic data to be generalized to both in- and out-of-domain settings. This property has been studied in (Petzka et al., 2021; Kaddour

et al., 2022), in the uni-level optimization. In this work, we forage this into the bilevel optimization framework by integrating Shaprness-Aware minimization. To jointly optimize the sharpness of the outer-loop and the distance between the trajectory w.r.t to the synthetic dataset, we maximize the objective function in the $\rho$ regime for the sharpness proxy approximation and then optimize the distance between trajectories according to the gradient computed on the local maximum for the dataset learning. This process is described as follows:

$$\min_{\phi} \max_{||\epsilon||_2 \leq \rho} \mathcal{L}(\theta^S(\phi + \epsilon)) = \frac{1}{\delta}||\theta^S_{t+N}(\phi + \epsilon) - \theta^E_{t+M}||_2^2 \quad \text{s.t.} \ \theta^S_{t+N}(\phi) = \Xi_N(\theta^S_t, \phi). \quad (4)$$

We define $F(\phi) = \mathcal{L}(\theta^S_{t+N}(\phi))$ to eliminate the effect of the inner-loop solution on the outer-loop loss value without losing generality. The perturbation vector, $\epsilon$, is computed through a closed-form solution derived through the first-order Taylor expansion of the objective function in Eq. 1.

$$\epsilon = \arg\max_{||\epsilon||_2 \leq \rho} \mathcal{L}(\theta^S(\phi + \epsilon)) = \arg\max_{||\epsilon||_2 \leq \rho} F(\phi + \epsilon) \approx \rho \frac{\nabla F(\phi)}{||\nabla F(\phi)||_2}. \quad (5)$$

The closed-form solution given in Eq. 5 can be interpreted as a one-step gradient ascent. However, this one-step gradient ascent may fail to reach the local maximum of the sharpness proxy, due to the high variance of the hypergradient. This phenomenon has also been observed by (Liu et al., 2022; Du et al., 2022) in the uni-level optimization and will aggravate in the complicated bilevel case (Abbas et al., 2022). To conduct accurate sharpness approximation, motivated by (Liu et al., 2022; Haruki et al., 2019; Wen et al., 2018; Duchi et al., 2012), we introduce fluctuation on the learnable dataset to smooth the landscape. To be more specific, each synthetic image indexed by $j$ is perturbed by a random noise sampled from a Gaussian distribution with a diagonal covariance matrix whose magnitude is proportional to the norm of the image $||\phi_j||$:

$$\phi_j = \phi_j + \phi_j^\Delta, \quad \phi_j^\Delta \sim \mathcal{N}(0, \gamma||\phi_j||_2),$$

where $\gamma$ is a tunable hyperparameter controlling the fluctuation strength. This process is conducted on the image independently in each one-step gradient ascent.

## 4.2 Efficient Sharpness-Aware Minimization in Bilevel Optimization

One can notice that a one-step update in the outer-loop requires twice hypergradient computation, namely one for the perturbation vector $\epsilon$ and the other for the real update gradient, $\nabla F(\phi)$. Directly computing those two gradients will double the computation cost in contrast with MTT and FTD instead of TESLA, which will be discussed later. To alleviate this problem, we proposed two approximation strategies, Truncated Unrolling Hypergradient (TUH) and Trajectory Reusing (TR) to reduce the computational overhead without harming the performance.

**Truncated Unrolling Hypergradient.** The long inner-loop horizon burdens the hypgradient estimation and introduces tremendous computational overhead. In our framework, the hypergradient for updating the learnable dataset is computed by differentiating through the unrolled computational graph of the inner-loop. This vanilla hypergradient computation scales the memory cost with the number of inner-loop iterations which is not feasible as condensing the complicated datasets requires long horizon inner-loops. Instead, we *truncate the backpropagation* by only differentiating through the last several steps of the inner-loop. This reduces both the memory and computational time. More percisely, the truncated hypergradient computation with $N$ step unrolling can be expressed as:

$$\frac{\partial F_\iota(\phi)}{\partial \phi} = \sum_{i=\iota}^{N} \frac{\partial \mathcal{L}(\theta(\phi))}{\partial \theta_N} \left( \prod_{i'=i}^{N} \frac{\partial \theta_{i'}}{\partial \theta_{i'-1}} \right) \frac{\partial \theta_i}{\partial \phi}, \quad (6)$$

where $\iota$ controls the number of truncated steps that $N - \iota$ steps of the inner-loop will be differentiated through. In addition, the risk of hypergradient exploding and vanishing caused by the ill-Jacobian $\frac{\partial \theta_i}{\partial \theta_{i-1}}$, which may happen in any inner-loop step, can be reduced. This mechanism can be easily implemented by releasing the computational graph while optimising the inner-loop and then creating the computational graph at a certain iteration with PyTorch-based pseudocode given in Appx. A.3.

We analyse the discrepancy between hypergradients computed by the truncated and untruncated computational graph in the setting where the synthetic trajectory is produced by optimizing from the initialisation $\theta_0^E$ until convergence.

**Theorem 4.1.** *Assmue $\mathcal{L}_{CE}$ is $K$-smooth, twice differentiable, and locally $J$-strongly convex in $\theta$ around $\{\theta_{\iota+1}, ..., \theta_N\}$. Let $\Xi(\theta, \phi) = \theta - \alpha \nabla \mathcal{L}_{CE}(\theta, \phi)$. For $\alpha \leq \frac{1}{K}$, then*

$$\left\| \frac{\partial F(\phi)}{\partial \phi} - \frac{\partial F_\iota(\phi)}{\partial \phi} \right\| \leq 2^\iota (1 - \alpha J)^{N-\iota+1} \left\| \frac{\partial \mathcal{L}(\theta(\phi))}{\partial \theta_N(\phi)} \right\| \max_{i \in \{0,...\iota\}} \left\| \frac{\partial \theta_i}{\partial \phi} \right\|,$$

*where $\frac{\partial F(\phi)}{\partial \phi}$ denotes the untruncated hypergradient.*

The Proposition 4.1 shows that the error of the truncated hypergradient decreases exponentially in $N - \iota + 1$ when $\theta$ converges to the neighbourhood of a local minimum in the inner-loop and the proofs in terms of both deterministic and stochastic inner loop are given in Appx. A.2.

**Trajectory Reusing.** The sharpness-aware minimization requires computing the gradient twice for sharpness proxy approximation and free parameter update, which means in bilevel optimization the inner-loop is required to unroll twice. This boosts the computational spending and slows down the training speed when inner-loops comprise long trajectories. To improve the efficiency of training, we propose to reuse the trajectory generated by the first round of inner-loop unrolling. We denote the trajectories generated by training on the perturbed dataset as $\hat{\theta}_i(\phi + \epsilon)$. Other than unrolling the entire second trajectory initialised by the expert segment, the training is initialised by the middle point, indexed by $\tau$, from the first trajectory $\hat{\theta}_\tau(\phi + \epsilon) := \theta_\tau(\phi)$. Note that the hypergradient for the dataset update is truncated implicitly since this hypergradient approximation will not consider the steps earlier than $\tau$ which is further constrained, $\tau \geq \iota$. Coupled with the same truncated strategy for the first round, the hypergradient in the second trajectory is computed as:

$$\frac{\partial F_{\tau,\epsilon}(\phi)}{\partial \phi} = \frac{\partial \mathcal{L}(\theta(\phi))}{\partial \theta_\tau} \frac{\partial \theta_\tau}{\partial \phi} = \sum_{i=\tau}^{N} \frac{\partial \mathcal{L}(\theta(\phi))}{\partial \theta_N} \left( \prod_{i'=i}^{N} \frac{\partial \theta_{i'}}{\partial \theta_{i'-1}} \right) \frac{\partial \theta_i}{\partial \phi} \Bigg|_{\phi=\phi+\epsilon, \ \hat{\theta}_\tau(\phi+\epsilon)=\theta_\tau(\phi)} \tag{7}$$

One may notice that the trajectory reusing strategy assumes the difference between two trajectories before step $\tau$ can be ignored. To rigorously study the effect of this assumption, we analyse the distance between $\theta_\tau(\phi)$ and $\theta_\tau(\phi + \epsilon)$. Similar to the Growth recursion lemma (Hardt et al., 2016) applied to upper-bound the difference between two weight points of two different trajectories trained by the dataset with only one data point difference. We develop the bound for the difference between two weight points at the same iteration of their trajectories generated by the datasets with and without perturbation below. The proof is provided in Appx.A.1.

**Theorem 4.2.** *Let $\mathcal{L}(\phi, \theta)$ be a function that is $\sigma$-smooth and continuous with respect to its arguments $\phi$ and $\theta$. Additionally, let the second-order derivatives $\nabla_\phi \nabla_\theta \mathcal{L}(\phi, \theta)$ be $\beta$-continuous. Consider two trajectories obtained by conducting gradient descent training on the datasets $\phi$ and $\phi + \epsilon$, respectively, with a carefully chosen learning rate $\alpha$ and identical initializations. After $\tau$ steps of training, let $\Delta \theta_\tau = \hat{\theta}_\tau(\phi + \epsilon) - \theta_\tau(\phi)$. Then, we have:*

$$\|\Delta \theta_\tau\| \leq \alpha \tau (2\sigma + \beta\rho).$$

This theorem tells us that the bound of the distance between those two end points is associated with the learning rate and the number of iterations. Thus, when the learning rate and $\tau$ are selected reasonably, $\theta_\tau(\phi)$ approximate $\hat{\theta}_\tau(\phi + \epsilon)$ properly. In addition, we set $\tau = \iota$ in our experiments to reduce the hyperparameter tuning efforts, even though tuning them separately may achieve better results. We compare the time and memory complexity of our method and Reverse Model Reverse Mode Differentiation (RMD) used in MTT (Cazenavette et al., 2022) and FTD (Du et al., 2023) in Table 1 to exhibit the efficiency provided by our method. In essence, SATM is designed to conduct efficient sharpness and trajectory distance minimization in the outer-loop of the bilevel optimization-based DC methods. The proposed approximation strategies, including THU and TR, are flexible enough to adapt to other advanced sharpness-aware optimizers such as ASAM (Kwon et al., 2021) and Vasson (Li & Giannakis, 2024).

## 5 EXPERIMENTS

We evaluate SATM on a variety of in-domain tasks, where the neural architecture and data distribution are consistent between training and testing. Meanwhile, we explore out-of-domain scenarios, including cross-architecture and cross-task settings, to demonstrate the generalization benefits achieved

Table 1: The computational complexity analysis for different trajectory matching-based algorithms in time and memory cost. $c$ is the time cost for computing $\Xi(\theta, \phi)$ with $\theta \in R^P$ and $\phi \in R^Q$. P and Q denote the dimensions of the base model and synthetic dataset.

| Methods | Time | Memory |
|---|---|---|
| MTT, FTD | $\mathcal{O}(cN)$ | $\mathcal{O}(PN)$ |
| TESLA | $\mathcal{O}(2cN)$ | $\mathcal{O}(P)$ |
| TUH + TR | $\mathcal{O}(cN + c\tau)$ | $\mathcal{O}(P(N - \iota))$ |

Table 2: Test Accuracy (%) Comparison of different image per category (IPC) setting on Cifar10, Cifar-100 and Tiny ImageNet: the models are trained on the synthetic dataset learned by MTT and our method independently and evaluated on the corresponding test set with real images. We cite the results of DC, DM and MMT from FTD (Du et al., 2023).

| Method | IPC | DC | DSA | DM | MTT | FTD | TESLA | MDC | Ours |
|---|---|---|---|---|---|---|---|---|---|
| Cifar-10 | 1 | $28.3_{\pm0.5}$ | $28.8_{\pm0.7}$ | $26.0_{\pm0.8}$ | $46.2_{\pm0.8}$ | $46.8_{\pm0.3}$ | $48.5_{\pm0.8}$ | $47.5_{\pm0.4}$ | $\mathbf{49.0}_{\pm0.3}$ |
| | 10 | $44.9_{\pm0.5}$ | $52.1_{\pm0.6}$ | $48.9_{\pm0.6}$ | $65.4_{\pm0.7}$ | $66.6_{\pm0.3}$ | $66.4_{\pm0.8}$ | $66.7_{\pm0.7}$ | $\mathbf{67.1}_{\pm0.3}$ |
| | 50 | $53.9_{\pm0.5}$ | $60.6_{\pm0.5}$ | $63.0_{\pm0.4}$ | $71.6_{\pm0.2}$ | $73.8_{\pm0.3}$ | $72.6_{\pm0.7}$ | $73.7_{\pm0.3}$ | $\mathbf{73.9}_{\pm0.2}$ |
| Cifar-100 | 1 | $12.8_{\pm0.3}$ | $13.9_{\pm0.3}$ | $11.4_{\pm0.3}$ | $24.3_{\pm0.3}$ | $25.2_{\pm0.2}$ | $24.8_{\pm0.4}$ | $25.9_{\pm0.2}$ | $\mathbf{26.1}_{\pm0.4}$ |
| | 10 | $25.2_{\pm0.3}$ | $32.3_{\pm0.3}$ | $29.7_{\pm0.3}$ | $39.7_{\pm0.4}$ | $43.4_{\pm0.3}$ | $41.7_{\pm0.3}$ | $42.7_{\pm0.6}$ | $\mathbf{45.2}_{\pm0.3}$ |
| | 50 | - | $42.8_{\pm0.4}$ | $43.6_{\pm0.4}$ | $47.7_{\pm0.2}$ | $50.7_{\pm0.3}$ | $47.9_{\pm0.3}$ | $49.6_{\pm0.4}$ | $\mathbf{53.2}_{\pm0.7}$ |
| TinyImageNet | 1 | - | - | $3.9_{\pm0.2}$ | $8.8_{\pm0.3}$ | $10.4_{\pm0.3}$ | $7.8_{\pm0.2}$ | $9.9_{\pm0.2}$ | $\mathbf{10.9}_{\pm0.2}$ |
| | 10 | - | - | $12.9_{\pm0.4}$ | $23.2_{\pm0.2}$ | $24.5_{\pm0.2}$ | $20.8_{\pm0.9}$ | $24.8_{\pm0.4}$ | $\mathbf{25.9}_{\pm0.4}$ |
| | 50 | - | - | $24.1_{\pm0.3}$ | $28.0_{\pm0.3}$ | $28.2_{\pm0.3}$ | $27.8_{\pm1.1}$ | $28.1_{\pm0.2}$ | $\mathbf{29.4}_{\pm0.3}$ |

through sharpness minimization. Complete experimental configurations, including dataset and architecture details, are provided in Appendix A.11.

**Popular Benchmark.** We compare our method against the other dataset condensation methods, such as DC (Zhao et al., 2021), DSA (Zhao & Bilen, 2021), DM (Zhao & Bilen, 2023), MTT (Cazenavette et al., 2022), FTD (Du et al., 2023), TESLA (Cui et al., 2023) and MDC (He et al., 2024). From the results in Table 2, one can observe that SATM outperforms the competitors on all the settings of the standard dataset condensation benchmarks with different IPCs while demonstrating the benefits of the flat convergence region. This benefit can be further observed in the high-resolution image condensation task in Table 3. Note that in our case, we merely build SATM up on Vanilla MMT (Cazenavette et al., 2022) without integrating the flat trajectory trick in FTD and the soft label in TESLA. Still, there are clear improvement margins over other trajectory-matching-based DC competitors.

**ImageNet Comparison with TESLA.** Due to the high memory cost of the trajectory matching-based dataset condensation methods, most existing works fail to distil synthetic datasets from ImageNet (Russakovsky et al., 2015). To achieve this, TESLA (Cui et al., 2023) trades off time complexity and performance to maintain a constant memory cost by unrolling its inner loop twice for one outer loop update. Our method also requires executing the inner-loop twice; however, it not only can achieve constant memory and lower time cost than TESLA but also achieves noticeable margin improvement on ImageNet in Table 4.

**Cross Architecture.** Whether the performance advantage conferred by flatness can generalize across architectures is a key practical question. To explore this, we evaluate the learned datasets on a variety of architectures. As shown in Table 5, the synthetic datasets generated by SATM for CIFAR-10 exhibit strong generalization to unseen architectures under both the IPC 10 and IPC 50 settings, outperforming those produced by MTT (Cazenavette et al., 2022) and FTD (Du et al., 2023). Notably, a dataset that performs well in the in-domain setting may not retain its effectiveness in the cross-architecture setting. For example, FTD performs comparably to SATM on CIFAR-10 under both IPC settings when evaluated on the ConvNet architecture used during dataset synthesis. However, the performance gap becomes significant when these datasets are applied to different architectures, highlighting the superior cross-architecture generalization of SATM.

**Continual Learning.** We expose the learned dataset to the task incremental setting, following the same protocol discussed in Gdumb (Prabhu et al., 2020). To be more specific, during the learning stage, the models encounter a sequence of data from different categories and lose access to the previous data after training. A limited memory budget is available to save historical training

Table 3: Test accuracy (%) comparison on the Subsets of ImageNet, including ImageNette, Image-Woor, ImageFruit, and ImageNeow with high-resolution ($128 \times 128$): All the synthetic datasets are learned and tested on ConvNet with 10 IPC.

|  | ImageNette | ImageWoof | ImageFruit | ImageMeow |
|---|---|---|---|---|
| MTT | $63.0_{\pm 1.3}$ | $35.8_{\pm 1.8}$ | $40.3_{\pm 1.3}$ | $40.4_{\pm 2.2}$ |
| FTD | $67.7_{\pm 0.7}$ | $38.8_{\pm 1.4}$ | $44.9_{\pm 1.5}$ | $43.3_{\pm 0.6}$ |
| Ours | $\mathbf{68.2}_{\pm 0.5}$ | $\mathbf{39.4}_{\pm 1.2}$ | $\mathbf{45.2}_{\pm 1.3}$ | $\mathbf{45.4}_{\pm 0.9}$ |
| All | $87.4_{\pm 1.0}$ | $67.0_{\pm 1.3}$ | $63.9_{\pm 2.0}$ | $66.7_{\pm 1.1}$ |

Table 4: Accuracy Comparison of TESLA and SATM across different IPCs on ImageNet-1K and average time cost (sce) comparison with 50 inner-loop iterations and 50 IPCs.

| Model/IPC | 1 | 10 | 50 | Time Cost |
|---|---|---|---|---|
| TESLA | $7.7_{\pm 0.2}$ | $17.8_{\pm 1.3}$ | $27.9_{\pm 1.2}$ | $46.8_{\pm 0.3}$ |
| SATM | $\mathbf{8.9}_{\pm 0.3}$ | $\mathbf{19.2}_{\pm 0.9}$ | $\mathbf{30.2}_{\pm 0.6}$ | $\mathbf{44.7}_{\pm 0.2}$ |

information to prevent catastrophic forgetting while adapting to new tasks. In Figure 1, we show that on every stage, our learned datasets outperform others in three settings: 5-task incremental with 50 images per category on Cifar10, 10-and 20-task incremental with 3 IPC on Tiny ImageNet, achieve outstanding generalization ability over different tasks.

**Computational Cost Comparison.** We compare the memory and time costs among MTT, TESLA, and SATM with cost quantity measured on NVIDIA A6000 GPUs. In our experiments, only one-third of the inner loop is retained to compute the hypergradients for sharpness approximation. Given the result in Table 6, our method significantly reduces memory consumption compared to MTT, enabling the dataset to be trained on a single A6000 GPU. Regarding time cost, SATM consistently outperforms the two inner-loop-based algorithms, TESLA, and more interestingly, SATM even consumes less time than MTT due to its algorithm requirements of retaining a full single inner-loop.

**Hypergradient Analysis.** To illustrate the effects of sharpness minimization on the synthetic dataset learning dynamic, we record the hypergradient norm of MTT and SATM and report their mean and variance over training iterations. Depicted in Fig 2, SATM has a smaller mean and variance than MTT on Cifar100 with 3 IPC and Tiny ImageNet 3IPC, indicating a stable learning process with fewer hypergradient spikes on SATM, reflecting the flatness of the loss landscape. Moreover, the decreasing trend of sharpness, measured by $\mathcal{L}(\phi + \epsilon) - \mathcal{L}(\phi)$, demonstrates the convergence towards the flat loss region. This explains why our method enjoys better generalization ability.

**Aligning with Curriculum Learning.** DATM (Guo et al., 2024) utilizes the difficulty of training trajectories to implement a curriculum learning-based DC algorithm and proposes to learn soft labels, unlike the research focusing on bilevel optimization efficiency, such as TESLA, FTD, and SATM. To explore whether SATM can be compatible with curriculum learning trajectories. We conduct experiments integrating DATM's easy-to-hard training protocol and the soft label alignment with SATM denoted as SATM-DA, yielding the positive results in Table 7.

Table 5: Test accuracy (%) comparison on Cifar10 with 10 and 50 images per class setting: the synthetic datasets by MTT, FTD, and our algorithm are learned on ConvNet and tested on AlexNet, VGG11, ResNet18, ResNet152, and ViT.

| Methods | IPC | ConvNet | AlexNet | VGG11 | ResNet18 | ResNet152 | ViT |
|---|---|---|---|---|---|---|---|
| MTT |  | $64.3_{\pm 0.7}$ | $34.2_{\pm 2.6}$ | $50.3_{\pm 0.8}$ | $46.4_{\pm 0.6}$ | $17.8_{\pm 1.4}$ | $34.9_{\pm 0.6}$ |
| FTD | 10 | $66.6_{\pm 0.4}$ | $36.5_{\pm 1.1}$ | $50.8_{\pm 0.3}$ | $46.2_{\pm 0.7}$ | $17.4_{\pm 1.2}$ | $35.2_{\pm 0.4}$ |
| Ours |  | $\mathbf{67.1}_{\pm 0.5}$ | $\mathbf{37.8}_{\pm 0.8}$ | $\mathbf{51.4}_{\pm 0.3}$ | $\mathbf{47.7}_{\pm 0.4}$ | $\mathbf{18.9}_{\pm 1.4}$ | $\mathbf{36.9}_{\pm 0.6}$ |
| MTT |  | $71.6_{\pm 0.2}$ | $48.2_{\pm 1.0}$ | $55.4_{\pm 0.8}$ | $61.9_{\pm 0.7}$ | $20.9_{\pm 1.6}$ | $47.7_{\pm 0.7}$ |
| FTD | 50 | $73.8_{\pm 0.2}$ | $53.8_{\pm 0.9}$ | $58.4_{\pm 1.6}$ | $65.7_{\pm 0.3}$ | $22.7_{\pm 1.2}$ | $50.1_{\pm 0.8}$ |
| Ours |  | $\mathbf{74.2}_{\pm 0.3}$ | $\mathbf{56.9}_{\pm 0.7}$ | $\mathbf{63.5}_{\pm 1.1}$ | $\mathbf{66.1}_{\pm 0.5}$ | $\mathbf{23.6}_{\pm 1.1}$ | $\mathbf{52.7}_{\pm 0.4}$ |

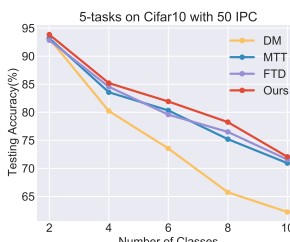 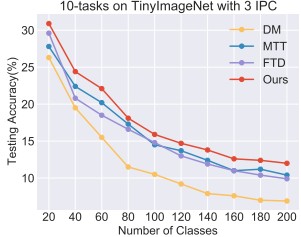 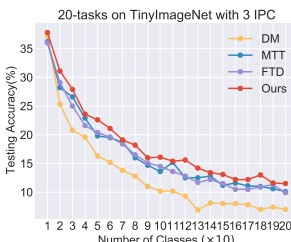

Figure 1: Test accuracy (%) comparison on continual learning. Left: 5-step class-incremental learning on Cifar10 50IPC, Middle: 10-step class-incremental learning on Tiny ImageNet 3IPC, Right: 20-step class-incremental learning on Tiny ImageNet 3IPC.

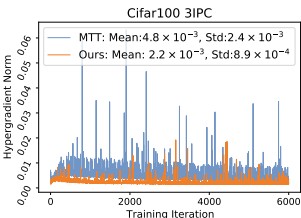 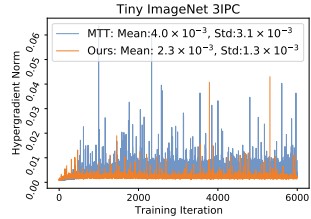 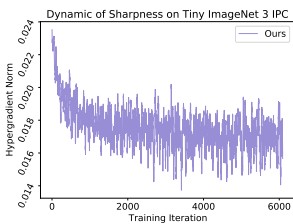

Figure 2: Sharpness analysis by visualisation. Hypergradient Norm comparison between MTT and SATM. Left: the hypergradient norm on Cifar100 with 10 IPC; Middle: the hypergradient norm on Tiny ImageNet with 3 IPC. Right: Sharpness dynamic on Tiny ImageNet with 3 IPC.

Table 6: GPU memory (GB) and runtime (sec) comparison among MTT, TESLA, and SATM on CIFAR100 and ImageNet-1K with results measured with a batch size of 100 and 50 inner-loop steps.

| Metric / Dataset | MTT | TESLA | SATM |
|---|---|---|---|
| CIFAR-100 | | | |
| Memory | $17.1_{\pm 0.1}$ | $3.6_{\pm 0.1}$ | $5.7_{\pm 0.1}$ |
| Runtime | $12.1_{\pm 0.6}$ | $15.3_{\pm 0.5}$ | $12.0_{\pm 0.5}$ |
| ImageNet-1K | | | |
| Memory | $80.5_{\pm 0.1}$ | $17.4_{\pm 0.1}$ | $26.5_{\pm 0.1}$ |
| Runtime | $45.9_{\pm 0.5}$ | $47.4_{\pm 0.7}$ | $45.4_{\pm 0.4}$ |

Table 7: Accuracy (%) Comparison of DATM, and SATM-DA across various IPCs, datasets, and configurations.

| IPC | DATM | SATM-DA |
|---|---|---|
| CIFAR-100 | | |
| 1 | $27.9_{\pm 0.2}$ | $\mathbf{28.2}_{\pm 0.8}$ |
| 10 | $47.2_{\pm 0.4}$ | $\mathbf{48.3}_{\pm 0.4}$ |
| 50 | $55.0_{\pm 0.2}$ | $\mathbf{55.7}_{\pm 0.3}$ |
| Tiny-ImageNet | | |
| 1 | $\mathbf{17.1}_{\pm 0.3}$ | $16.4_{\pm 0.4}$ |
| 10 | $31.1_{\pm 0.3}$ | $\mathbf{32.3}_{\pm 0.6}$ |
| 50 | $39.7_{\pm 0.3}$ | $\mathbf{40.2}_{\pm 0.7}$ |

# 6 CONCLUSION

In this work, we explore the generalization ability of condensed datasets produced by the training trajectory-matching family. We propose Sharpness-Aware Trajectory Matching (SATM), to optimize both the sharpness of the loss landscape and trajectory distances while achieving low learning cost. To improve such efficiency, the hypergradient estimation on the long inner loop and out loop sharpness costs is reduced through two theoretically guided strategies. Our approach improves generalization across in- and out-of-domain tasks, such as cross-architecture and continual learning, and can be further successfully deployed to the challenging ImageNet-1K task with clear improvement on both generalization performance and computational overhead. Additionally, SATM serves as a "plug-and-play" model for other methods, especially the curriculum learning-based method, resulting in further improvement. Future research could explore advanced gradient estimation techniques, such as implicit gradient, to enhance computational efficiency and reduce approximation error.

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

# A APPENDIX

## A.1 PROOF FOR THEOREM 4.2

**Theorem 4.2.** *Let $\mathcal{L}(\phi, \theta)$ be a function that is $\sigma$-smooth and continuous with respect to its arguments $\phi$ and $\theta$. Additionally, let the second-order derivatives $\nabla_\phi \nabla_\theta \mathcal{L}(\phi, \theta)$ be $\beta$-continuous. Consider two trajectories obtained by conducting gradient descent training on the datasets $\phi$ and $\phi + \epsilon$, respectively, with a carefully chosen learning rate $\alpha$ and identical initializations. After $\tau$ steps of training, let $\Delta\theta_\tau = \hat\theta_\tau(\phi + \epsilon) - \theta_\tau(\phi)$. Then, we have:*

$$\|\Delta\theta_\tau\| \leq \alpha\tau(2\sigma + \beta\rho).$$

*Proof.* Let:

$$\hat\theta_\tau = \theta_0 - \alpha \sum_i^\tau \nabla\mathcal{L}(\phi + \epsilon, \hat\theta_i)$$

$$\theta_\tau = \theta_0 - \alpha \sum_i^\tau \nabla\mathcal{L}(\phi, \theta_i)$$

then after $\tau$ step iterations, the difference between $\theta_\tau$ and $\hat\theta_\tau$ is

$$\|\Delta\theta_\tau\| = \left\|\hat\theta_\tau - \theta_\tau\right\| = \left\|-\alpha \sum_i^\tau (\nabla\mathcal{L}(\phi + \epsilon, \hat\theta_i) - \nabla\mathcal{L}(\phi, \theta_i))\right\|$$

$$= \alpha \left\|\sum_i^\tau (\nabla\mathcal{L}(\phi + \epsilon, \hat\theta_i) - \nabla\mathcal{L}(\phi, \theta_i))\right\|.$$

We compute the gradient difference:

$$||\nabla\mathcal{L}(\phi + \epsilon, \hat\theta_i) - \nabla\mathcal{L}(\phi, \theta_i)||$$
$$\approx ||\nabla\mathcal{L}(\phi, \hat\theta_i) + \nabla_\phi \nabla_\theta \mathcal{L}(\phi, \hat\theta_i) \cdot \epsilon - \nabla\mathcal{L}(\phi, \theta_i)||$$
$$\leq ||\nabla\mathcal{L}(\phi, \hat\theta_i) - \nabla\mathcal{L}(\phi, \theta_i)|| + ||\nabla_\phi \nabla_\theta \mathcal{L}(\phi, \hat\theta_i) \cdot \epsilon||$$
$$\leq 2\sigma + ||\nabla_\phi \nabla_\theta \mathcal{L}(\phi, \hat\theta_i)|| ||\epsilon||.$$

With $\nabla_\phi \nabla_\theta \mathcal{L}(\phi, \hat\theta_i)$ is $\beta$ smooth and $||\epsilon|| = \rho$ :

$$||\nabla\mathcal{L}(\phi + \epsilon, \hat\theta_i) - \nabla\mathcal{L}(\phi, \theta_i)|| \leq 2\sigma + \beta\rho.$$

Then:

$$\|\Delta\theta_\tau\| \leq \alpha\tau(2\sigma + \beta\rho)$$

$\square$

We extend the analysis result developed for gradient descent to the stochastic gradient descent setting, assuming the unbiased stochastic gradient with bounded variance.

**Theorem A.1.** *Let $\mathcal{L}(\phi, \theta)$ be a function that is $\sigma$-smooth and continuous with respect to its arguments $\phi$ and $\theta$. Additionally, let the second-order derivatives $\nabla_\phi \nabla_\theta \mathcal{L}(\phi, \theta)$ be $\beta$-continuous. Consider two trajectories obtained by conducting stochastic gradient descent training on the datasets $\phi$ and $\phi + \epsilon$, respectively, with a carefully chosen learning rate $\alpha$ and identical initialization. During the optimization process, only the unbiased stochastic gradient with bounded variance, $\nabla\tilde{\mathcal{L}}(\phi, \theta_i)$, can be observed, which leads to $\mathbb{E}[\nabla\tilde{\mathcal{L}}(\phi, \theta)] = \nabla\mathcal{L}(\phi, \theta)$ and $\mathbb{E}[\|\nabla\tilde{\mathcal{L}}(\phi, \theta) - \nabla\mathcal{L}(\phi, \theta)\|^2] \leq \nu^2$. After $\tau$ steps of training, let $\Delta\theta_\tau = \hat\theta_\tau(\phi + \epsilon) - \theta_\tau(\phi)$. Then, we have:*

$$\|\Delta\theta_\tau\| \leq \alpha\tau(2\sigma + \beta\rho + \sqrt{2}\nu).$$

*Proof.* Let:

$$\hat{\theta}_\tau = \theta_0 - \alpha \sum_i^\tau \nabla \tilde{\mathcal{L}}(\phi + \epsilon, \hat{\theta}_i),$$

$$\theta_\tau = \theta_0 - \alpha \sum_i^\tau \nabla \tilde{\mathcal{L}}(\phi, \theta_i)$$

where

$$\tilde{\mathcal{L}}(\phi + \epsilon, \hat{\theta}_i) = \mathcal{L}(\phi + \epsilon, \hat{\theta}_i) + \hat{\xi}_i$$
$$\tilde{\mathcal{L}}(\phi, \theta_i) = \mathcal{L}(\phi, \theta_i) + \xi_i$$

with $\mathbb{E}[\xi_i] = \mathbb{E}[\hat{\xi}_i] = 0$ and $\mathbb{E}[\|\xi_i\|^2], \mathbb{E}[\|\hat{\xi}_i\|^2] \leq \nu^2$ through the unbiased unbound variance assumption.

Then after $\tau$ step iterations, the difference between $\theta_\tau$ and $\hat{\theta}_\tau$ is

$$\|\Delta\theta_\tau\| = \left\|\hat{\theta}_\tau - \theta_\tau\right\| = \left\|-\alpha \sum_i^\tau (\nabla \tilde{\mathcal{L}}(\phi + \epsilon, \hat{\theta}_i) - \nabla \tilde{\mathcal{L}}(\phi, \theta_i))\right\|$$

$$= \alpha \left\|\sum_i^\tau (\nabla \tilde{\mathcal{L}}(\phi + \epsilon, \hat{\theta}_i) - \nabla \tilde{\mathcal{L}}(\phi, \theta_i))\right\|.$$

Then for each stochastic step difference we have:

$$\|\nabla \tilde{\mathcal{L}}(\phi, \theta_i) - \nabla \tilde{\mathcal{L}}(\phi + \epsilon, \hat{\theta}_i)\| = \|\nabla \mathcal{L}(\phi, \theta_i) + \xi_i - \nabla \mathcal{L}(\phi + \epsilon, \hat{\theta}_i) - \hat{\xi}_i\|$$

$$\leq \|\nabla \mathcal{L}(\phi, \theta_i) - \nabla \mathcal{L}(\phi + \epsilon, \hat{\theta}_i)\| + \|\xi_i - \hat{\xi}_i\|.$$

We bound the difference for the first term:

$$\|\nabla \mathcal{L}(\phi + \epsilon, \hat{\theta}_i) - \nabla \mathcal{L}(\phi, \theta_i)\|$$

$$\approx \|\nabla \mathcal{L}(\phi, \hat{\theta}_i) + \nabla_\phi \nabla_\theta \mathcal{L}(\phi, \hat{\theta}_i) \cdot \epsilon - \nabla \mathcal{L}(\phi, \theta_i)\|$$

$$\leq \|\nabla \mathcal{L}(\phi, \hat{\theta}_i) - \nabla \mathcal{L}(\phi, \theta_i)\| + \|\nabla_\phi \nabla_\theta \mathcal{L}(\phi, \hat{\theta}_i) \cdot \epsilon\|$$

$$\leq 2\sigma + \|\nabla_\phi \nabla_\theta \mathcal{L}(\phi, \hat{\theta}_i)\|\|\epsilon\|.$$

With $\nabla_\phi \nabla_\theta \mathcal{L}(\phi, \hat{\theta}_i)$ is $\beta$ smooth and $\|\epsilon\| = \rho$, we have:

$$\|\nabla \mathcal{L}(\phi + \epsilon, \hat{\theta}_i) - \nabla \mathcal{L}(\phi, \theta_i)\|_2 \leq 2\sigma + \beta\rho.$$

Then based on $\|\xi_i - \hat{\xi}_i\| \leq \sqrt{2}\nu$ and sum $\tau$ steps:

$$\|\Delta\theta_\tau\| \leq \alpha\tau(2\sigma + \beta\rho + \sqrt{2}\nu)$$

$\square$

## A.2 PROOF OF PROPOSITION 4.1

**Theorem 4.1.** *Assmue $\mathcal{L}_{CE}$ is $K$-smooth, twice differentiable, and locally $J$-strongly convex in $\theta$ around $\{\theta_{\iota+1}, ..., \theta_N\}$. Let $\Xi(\theta, \phi) = \theta - \alpha \nabla \mathcal{L}_{CE}(\theta, \phi)$. For $\alpha \leq \frac{1}{K}$, then*

$$\left\|\frac{\partial F(\phi)}{\partial \phi} - \frac{\partial F_\iota(\phi)}{\partial \phi}\right\| \leq 2^\iota (1 - \alpha J)^{N-\iota+1} \left\|\frac{\partial \mathcal{L}(\theta(\phi))}{\partial \theta_N(\phi)}\right\| \max_{i \in \{0, ...\iota\}} \left\|\frac{\partial \theta_i}{\partial \phi}\right\|,$$

*where $\frac{\partial F(\phi)}{\partial \phi}$ denotes the untruncated hypergradient.*

*Proof.* Let

$$A_{i+1} = \frac{\partial \theta_{i+1}}{\partial \theta_i}, B_{i+1} = \frac{\partial \theta_{i+1}}{\partial \phi}$$

then

$$\frac{\partial F(\phi)}{\partial \phi} = \frac{\partial \mathcal{L}(\theta(\phi))}{\partial \phi} + \sum_{i=0}^{N} B_i A_{i+1} \cdots A_N \frac{\partial \mathcal{L}(\theta(\phi))}{\partial \theta_N(\phi)}$$

Let $e_\iota = \frac{\partial F(\phi)}{\partial \phi} - \frac{\partial F_\iota(\phi)}{\partial \phi}$,

$$e_\iota = \left( \sum_{i=0}^{\iota} B_i A_{i+1} \cdots A_\iota \right) A_{\iota+1} \cdots A_N \frac{\partial \mathcal{L}(\theta(\phi))}{\partial \theta_N(\phi)}$$

Given $\mathcal{L}_{CE}$ is locally $J$-strongly convex with respect to $\theta$ in the neighborhood of $\{\theta_{\iota+1}, \ldots, \theta_N\}$,

$$\|e_\iota\| \leq \left\| \sum_{i=0}^{\iota} B_i A_{i+1} \cdots A_\iota \right\| \left\| A_{\iota+1} \cdots A_N \frac{\partial \mathcal{L}(\theta(\phi))}{\partial \theta_N(\phi)} \right\|$$

$$\leq (1 - \alpha J)^{N-\iota+1} \left\| \frac{\partial \mathcal{L}(\theta(\phi))}{\partial \theta_N(\phi)} \right\| \left\| \sum_{i=0}^{\iota} B_i A_{i+1} \cdots A_\iota \right\|$$

In the worst case, when $\mathcal{L}_{CE}$ is $K$-smooth but nonconvex, then if the smallest eigenvalue of $\frac{\partial^2 \mathcal{L}_{CE}(\theta,\phi)}{\partial \theta \, \partial \theta}$ is $-K$, then $\|A_i\| = 1 + \alpha K \leq 2$ for $i = 0, \ldots, \iota$. $\qquad \square$

### A.3 Pytorch Based Pseudocode for Truncated Unrolling Hypergradient

---
**Algorithm 2** Trucated hypergradient computation

---
```
stop gradient:
```
**for** $i = 1, \ldots, \iota$ **do**
$\quad \theta_i = \theta_{i-1} - \alpha * \text{torch.grad}(\mathcal{L}_{CE}(\theta, \phi), \theta)$
**end for**
```
with gradient:
```
**for** $i = 1, \ldots, N - \iota$ **do**
$\quad \theta_i = \theta_{i-1} - \alpha * \text{torch.grad}(\mathcal{L}_{CE}(\theta, \phi), \theta, \text{retain\_graph} = \text{True}, \text{create\_graph} = \text{True})$
**end for**
**Return:** $\theta_N(\phi)$

---

### A.4 Learning-Rate Learning with First Order Derivative:

Adapting the inner loop learning rate, $\alpha$, to the different stages of dataset learning determines the performance of the learned dataset (Cazenavette et al., 2022). The automatic adaptation is achieved by modifying the learning rate by the hypergradient of the dataset learning objective function, $\frac{\partial \mathcal{L}(\phi)}{\partial \alpha}$. This hypergradient can be computed jointly with the hypergradient for the dataset learning, which is cumbersome in practice. To mitigate this burden, we derive an analytic solution for inner loop learning rate updating:

$$\alpha = \alpha - \lambda \frac{\partial \mathcal{L}(\theta_N(\phi))}{\partial \theta_N} \cdot \left( - \sum_{i=0}^{N-1} \frac{\partial \mathcal{L}_{CE}(\theta_i, \phi)}{\partial \theta_i} \right), \tag{8}$$

where $\lambda$ indicates the learning rate for the learning rate learning. This closed-form solution only aggregates the gradient of each step which only requires first-order derivative computation. We compare the learning rate learning dynamics produced by first-order (our method) and second-order derivatives, demonstrating limited differences between those two methods. The derivation and the details of experiments are given in Appx. A.5.

### A.5 The Derivation of Learning Rate Learning with First Order Derivative

In this section, we provide the derivation of the hypergradient calculation for learning rate $\alpha$ and the visual comparison of the learning rate learning dynamics generated by the first-order and second-order

methods. Given the outer-loop objective, $\mathcal{L}(\theta(\phi))$, and the inner-loop object $\mathcal{L}_{CE}(\theta_i, \phi)$ with $N$ iteration unrolling, the computation can be dedicated by:

$$
\begin{aligned}
\frac{\partial \mathcal{L}(\theta_N(\phi))}{\partial \alpha} &= \frac{\partial \mathcal{L}(\theta_N(\phi))}{\partial \theta_N} \cdot \frac{\partial(\theta_N, \phi)}{\partial \alpha} \\
&= \frac{\partial \mathcal{L}(\theta_N(\phi))}{\partial \theta_N} \cdot \frac{\partial \Xi(\theta_{N-1}, \phi)}{\partial \alpha} \\
&= \frac{\partial \mathcal{L}(\theta_N(\phi))}{\partial \theta_N} \cdot \frac{\partial}{\partial \alpha}\left( \theta_{N-1} - \alpha \frac{\partial \mathcal{L}_{CE}(\theta_{N-1}, \phi)}{\partial \theta_{N-1}} \right) \\
&= \frac{\partial \mathcal{L}(\theta_N(\phi))}{\partial \theta_N} \cdot \left( \frac{\partial \theta_{N-1}}{\partial \alpha} - \frac{\partial \mathcal{L}_{CE}(\theta_{N-1}, \phi)}{\partial \theta_{N-1}} \right) \\
&\quad \text{we treat } \frac{\partial \mathcal{L}_{CE}(\theta_{N-1}, \phi)}{\partial \theta_{N-1}} \text{ as a constant w.r.t. } \alpha \\
&= \frac{\partial \mathcal{L}(\theta_N(\phi))}{\partial \theta_N} \cdot \left( \frac{\partial}{\partial \alpha}\Xi(\theta_{N-2}, \phi) - \frac{\partial \mathcal{L}_{CE}(\theta_{N-1}, \phi)}{\partial \theta_{N-1}} \right) \\
&= \frac{\partial \mathcal{L}(\theta_N(\phi))}{\partial \theta_N} \cdot \left( -\sum_{i=0}^{N-1} \frac{\partial \mathcal{L}_{CE}(\theta_i, \phi)}{\partial \theta_i} \right)
\end{aligned}
$$

We compare the learning rate learning dynamics produced by first-order (our method) and second-order derivatives, demonstrating limited differences between those two methods. The visualisation comparison of the learning rate learning dynamic produced by the first and second-order derivative is illustrated in Fig. 3. As can be noticed, two inner-loop trajectories in the sharpness aware setting are capable of this Eq. 8. We chose the first in our experiments due to the implementation simplicity without causing any significant performance differences.

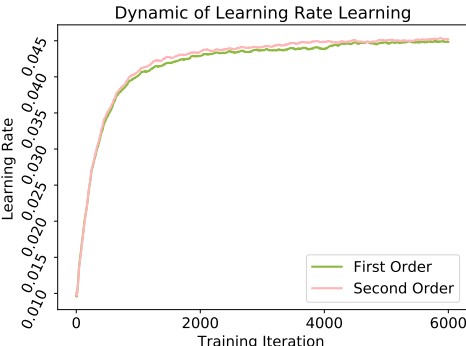

Figure 3: The comparison of the learning dynamics of learning rate learning with first and second order differentiation when condensing on the Cifar100-10IPC setting.

### A.6 COMPUTATIONAL RESOURCE

We conduct all our experiments on two TESLA V100-32GB GPUs with Intel(R) Xeon(R) W-2245 CPU @ 3.90GHz and one A100-40GB GPU with Intel(R) Xeon(R) Gold 5118 CPU @ 2.30GHz, which are on different servers. Thus, we cannot run the full batch of synthetic dataset learning as the same as other trajectory matching-based methods when the inner-loop trajectories contain many unrolling iterations. Those cases include Cifar100-10IPC, Cifar100-50IPC, and Tiny ImageNet 1IPC. In our case, stochastic gradient descent with mini-batch is utilised in the outer-loop instead.

## A.7 Flat Inner-loop Study

SATM is developed based on MTT without incorporating the components introduced in FTD (Du et al., 2023), particularly the expert trajectories generated by sharpness-aware optimizers such as GSAM. However, understanding whether SATM can be compatible with advanced expert trajectories is desirable to study. Therefore, we follow the expert trajectory generation protocol and execute SATM on the flat expert trajectories with the results in Table 8. It can be observed that the inclusion of a flat inner-loop leads to clear improvements in SATM-FI compared to both standard SATM and FTD. Furthermore, the authors of FTD noted the limited performance contribution of EMA, which was originally intended to guide the synthetic dataset toward convergence on a flat loss landscape. SATM addresses this limitation and effectively demonstrates the benefits of leveraging flatness for improved generalization.

|  | IPC | MTT | FTD | SATM | SATM-FI |
|---|---|---|---|---|---|
| CIFAR-10 | 1 | 46.2±0.8 | 46.8±0.3 | **49.0**±0.3 | 48.7±0.4 |
|  | 10 | 65.4±0.7 | 66.6±0.3 | 67.1±0.4 | **67.9**±0.3 |
|  | 50 | 71.6±0.2 | 73.8±0.2 | 73.9±0.2 | **74.2**±0.4 |
| CIFAR-100 | 1 | 24.3±0.3 | 25.2±0.2 | 26.1±0.4 | **26.6**±0.5 |
|  | 10 | 39.7±0.4 | 43.4±0.3 | 43.1±0.5 | **43.9**±0.7 |
|  | 50 | 47.7±0.2 | 50.7±0.3 | 53.2±0.7 | **54.4**±0.5 |
| Tiny-ImageNet | 1 | 8.8±0.3 | 10.4±0.3 | 10.9±0.2 | **11.7**±0.4 |
|  | 10 | 23.2±0.1 | 24.5±0.2 | 25.4±0.4 | **25.6**±0.6 |

Table 8: Accuracy (%) Comparison of MTT, FTD, SATM, and SATM-FI across different datasets and configurations.

## A.8 Compatibility with Advanced Sharpness-Aware optimizers.

We study the compatibility of the proposed hypergradient approximation method on other sharpness minimization-based methods, including EMA, SAM (Foret et al., 2020), GSAM (Zhuang et al., 2021), ASAM (Kwon et al., 2021) and Vasso (Li & Giannakis, 2024) with our loss landscape smoothing mechanism removed. For a fair comparison, the hyperparameters of each method are properly tuned for the adaptation to all the task,s including Cifar100 with 1 IPC and Tiny ImageNet with 3 IPC. We repeat each method 5 times and report the mean and variance in Table A.8. The results imply that all the sharpness methods consistently improve MTT (Cazenavette et al., 2022), which justifies the benefit of sharpness minimization. However, the competitors all fail to defeat our method due to the failure to accurately compute the sharpness proxy. Moreover, EMA, equivalent to FTD without Sharpness-aware minimizers to generate expert trajectories, gains minimal improvement.

| Dataset (IPC) | MTT | EMA | SAM | GSAM | ASAM | Vasso | SATM |
|---|---|---|---|---|---|---|---|
| Cifar100 (1) | 24.3±0.4 | 24.7±0.2 | 25.7±0.3 | 25.9±0.3 | 25.7 ± 0.3 | 25.9±0.2 | **26.1**±0.3 |
| Tiny ImageNet (3) | 10.5±0.3 | 10.9±0.3 | 12.3±0.2 | 13.1±0.2 | 12.8 ± 0.4 | 12.2±0.2 | **13.6**±0.2 |

Table 9: Test Accuracy (%) Comparison with the advanced sharpness aware minimization methods including EMA, SAM, GSAM, ASAM and Vasso with the same expert trajectories as MTT.

## A.9 Limitations and Future Works

In this work, we explore the generalization ability of condensed datasets produced by training trajectory-matching-based algorithms via jointly optimizing the sharpness and the distance between real and synthetic trajectories. We propose Sharpness-Aware Trajectory Matching (SATM) to reduce the computational cost caused by the long horizon inner-loop and the mini-max optimization for the sharpness minimization through the proposed hypergradient approximation strategies. Those strategies have clear theoretical motivation, limited error in practice, and a framework flexible enough to adapt to other sharpness-aware based algorithms. The improvement of the generalization is observed in a variety of in- and out-of-domain tasks such as cross-architecture and cross-task

(continual learning) with a comprehensive analysis of the algorithm's sharpness properties on the training dynamics.

Despite the superior performance of SATM, we observed that the proposed algorithm serves as a "plug-and-play" model for other dataset condensation methods and more broadly, for various bilevel optimization applications, such as loss function learning and optimizer learning. However, these possibilities are not explored in this work, and we leave them to future work. Moreover, beyond focusing on reusing the trajectory to enhance training efficiency in reaching flat regions, future research could be in advanced gradient estimation directions, such as implicit gradients, showing promise for managing long-horizon inner-loops and avoiding second-order unrolling. This could eliminate the entire second trajectory, resulting in higher computational efficiency and less approximation error.

## A.10 TRUNCATED STEP STUDY

We study the effects of the number of inner-loop steps remaining for the hypergradient estimation on the model performance. Table 10 details the settings, including the dataset, the number of images per category (IPC), and the inner-loop steps $N$. To analyze such effects, we retained the last $\frac{1}{k}$ steps, where $k = 2, 3, 4, 5, 6$, of the total inner-loop steps. The operation $int(\frac{N}{k})$ is used to determine the remaining inner-loop steps. From the results illustrate it can be noticed that performance improves as the number of truncated iterations decreases and converges once the differentiation steps reach a certain threshold.

Table 10: Accuracy (%) change along with the truncated inner-loop step change. We conduct SATM on CIFAR-10 to learn 1 image per category while running 50 iterations for the inner-loop and 80 iterations for the CIFAR-100 with 50IPC.

| Configuration | $\frac{1}{6}$ | $\frac{1}{5}$ | $\frac{1}{4}$ | $\frac{1}{3}$ | $\frac{1}{2}$ |
|---|---|---|---|---|---|
| CIFAR-10 | 45.2 | 48.8 | 47.5 | 49.0 | 49.2 |
| CIFAR-100 | 26.4 | 36.4 | 51.7 | 52.1 | 53.2 |

## A.11 EXPERIMENT SETTING DETAILS

We conduct experiments on four main image datasets, Cifar10 (Krizhevsky et al., 2009), Cifar100 (Krizhevsky et al., 2009), TinyImageNet (Le & Yang, 2015) and ImageNet (Russakovsky et al., 2015). Cifar10 categorises 50,000 images with the size $32 \times 32$ into 10 classes while Cifar100 further categorises each of those 10 classes into 10 fine-grained subcategories. TinyImageNet comprises 100,000 images distributed across 200 categories, each category consisting of 500 images resized to dimensions of $64 \times 64$. We further evaluate SATM on the subset of ImageNet, namely ImageNette, Image Woof, ImageFruit and ImageMeow with each set containing 10 different categories of $128 \times 128$ images and the whole ImageNet following the protocol from TESLA (Cui et al., 2023).

We evaluate our methods on four main image datasets, Cifar10 (Krizhevsky et al., 2009), Cifar100 (Krizhevsky et al., 2009), TinyImageNet (Le & Yang, 2015) and ImageNet-1K (Russakovsky et al., 2015). The expert trajectories for Cifar10 and Cifar100 are trained with 3-layer ConvNet and collected after each epoch with the initialisation, and those for TinyImageNet and ImageNet are trained with 4-layer and 5-layer ConvNet Gidaris & Komodakis (2018) respectively. In the in-domain setting, the synthetic datasets are learned and evaluated on the same architectures while in the out-of-domain settings, the learned synthetic datasets are deployed to train different architectures, such as AlexNet (Krizhevsky et al., 2012), VGG11 (Simonyan & Zisserman, 2014) and ResNet18 (He et al., 2016), which is novel to the synthetic datasets. The trained neural networks are evaluated on the real test sets for generalization ability comparison of the synthetic datasets.

## A.12 HYPERPARAMETERS AND EXPERIMENT DETAILS

The hyperparameters used for condensing datasets in all the settings are given in Tab 11 with ConvNet (Gidaris & Komodakis, 2018) applied to construct the training trajectories.

| Dataset | Model | IPC | Synthetic Steps (N) | Expert Epochs (M) | Max Start Epoch (T) | Synthetic Batch Size | ZCA | Learning Rate (Images) | Learning Rate (Step size) |
|---|---|---|---|---|---|---|---|---|---|
| CIFAR-10 | ConvNetD3 | 1 | 50 | 2 | 2 | - | Y | 1000 | $1 \times 10^{-6}$ |
| | | 3 | 50 | 2 | 2 | - | Y | 100 | $1 \times 10^{-5}$ |
| | | 10 | 30 | 2 | 20 | - | Y | 50 | $1 \times 10^{-5}$ |
| | | 50 | 30 | 2 | 40 | - | Y | 100 | $1 \times 10^{-5}$ |
| CIFAR-100 | ConvNetD3 | 1 | 40 | 3 | 20 | - | Y | 500 | $1 \times 10^{-5}$ |
| | | 3 | 45 | 3 | 20 | - | Y | 1000 | $5 \times 10^{-5}$ |
| | | 10 | 20 | 2 | 20 | 500 | Y | 1000 | $1 \times 10^{-5}$ |
| | | 50 | 80 | 2 | 40 | 500 | Y | 1000 | $1 \times 10^{-5}$ |
| Tiny ImageNet | ConvNetD4 | 1 | 30 | 2 | 10 | 200 | Y | 1000 | $1 \times 10^{-4}$ |
| | | 3 | 30 | 2 | 15 | 200 | Y | 1000 | $1 \times 10^{-4}$ |
| | | 10 | 20 | 2 | 40 | 200 | Y | 10000 | $1 \times 10^{-4}$ |

Table 11: Hyper-parameters used for our SATM. A synthetic batch size of "-" represents that a full batch set is used in each outer-loop iteration. ConvNetD3 and ConvNet4D denote the 3-layer and 4-layer ConvNet (Gidaris & Komodakis, 2018), respectively. In all the settings, ZCA whitening (Nguyen et al., 2021b;a) is applied.

## A.13 THE USE OF LARGE LANGUAGE MODELS (LLMs)

We claim that in this work, we only use LLMs for polishing the writing, such as checking the spelling and grammatical mistakes. LLMs are not used for our algorithm development, coding, and other essential contributions.

## A.14 ILLUSTRATION FOR THE SYNTHETIC IMAGES

We visualise the learned synthetic datasets on Cifar10, Cifar100, and Tiny ImageNet in this section.

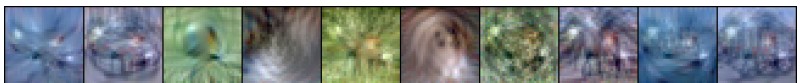

Figure 4: Cifar10 with 1IPC

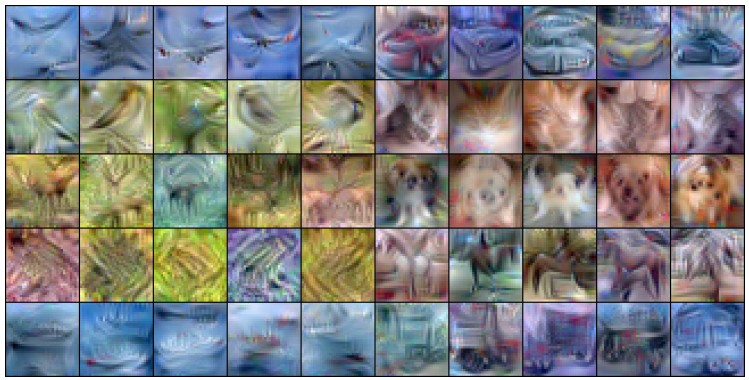

Figure 5: Cifar10 with 3IPC

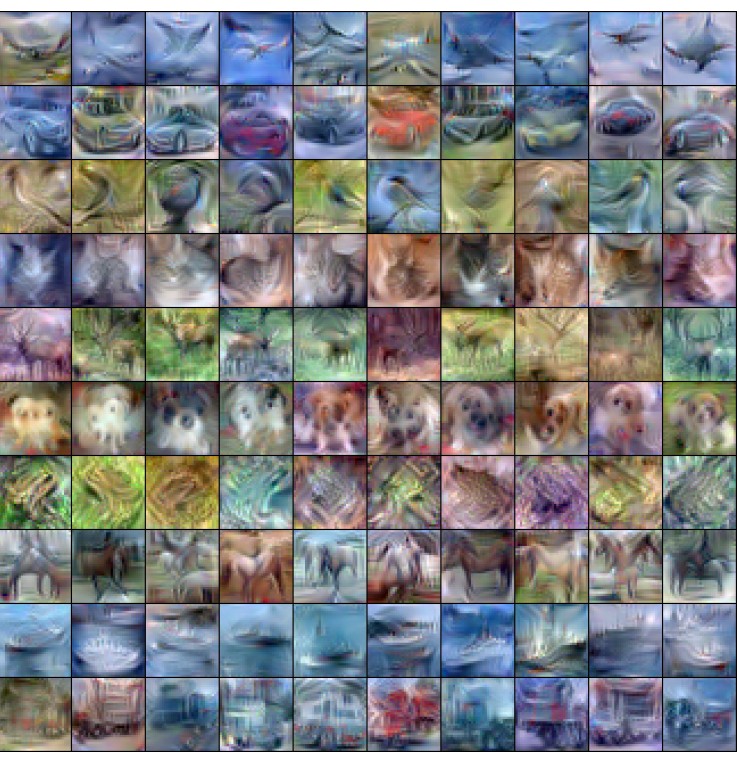

Figure 6: Cifar10 with 10IPC

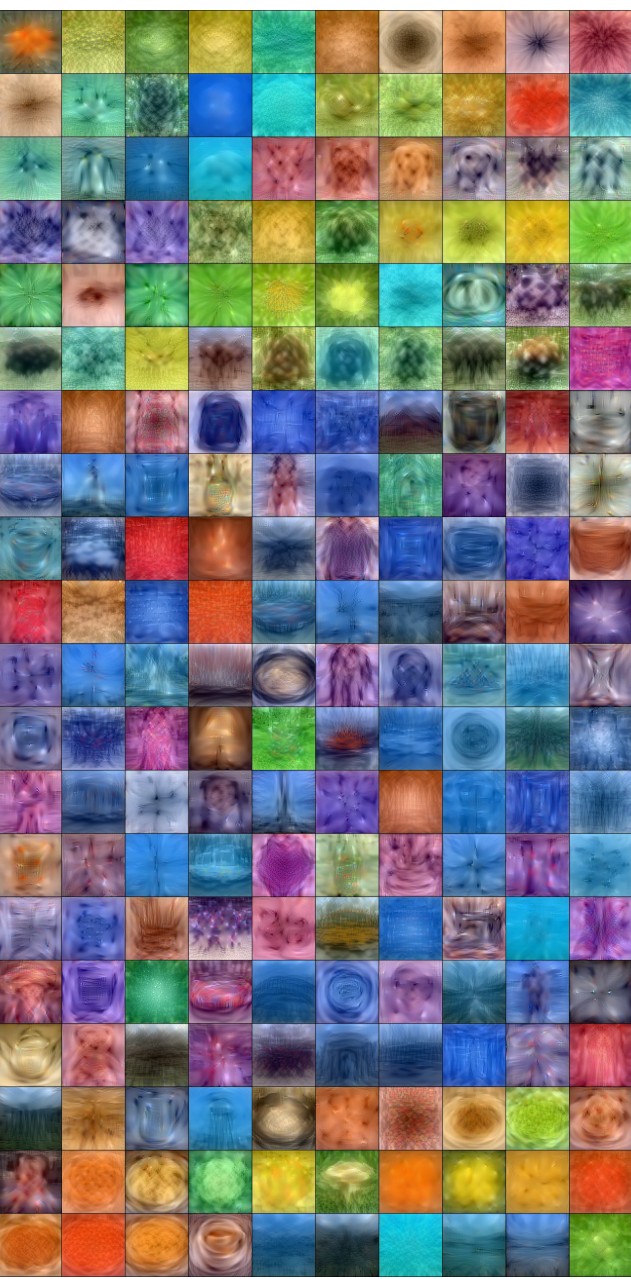

Figure 7: Cifar100 with 1IPC

