# OpenReview forum: "Enhancing Generalization via Sharpness-Aware Trajectory Matching for Dataset Condensation"
_ICLR.cc/2026/Conference — Submitted to ICLR 2026_

### Official Review · Reviewer_TTrg · 2025-10-18

**Soundness:** 3
**Presentation:** 3
**Contribution:** 2
**Rating:** 4
**Confidence:** 3

**Summary:**

This paper introduced the sharpness-aware minimizer into the bilevel-optimization-based dataset condensation to address the optimization issue from the complicated loss landscape. To reduce the computational overhead of the added optimization, two strategies are proposed to approximate the hypergradient calculation. Experiments across various settings show the effectiveness of the proposed solution to optimizes the bilevel objective function and sharpness simultaneously.

**Strengths:**

1. The optimization issue of the bilevel problem in dataset condensation is very important. This work studied this problem from the less-explored loss landscape view.
2. The proposed strategies for mitigating the doubled computational cost of SAM are mathematically sound and practical.
3. The proposed solution achieves consistent improvements over state-of-the-art trajectory-matching competitors across standard benchmarks.

**Weaknesses:**

1. The novelty of the proposed solution is not large. The introduction of SAM into dataset condensation  in Section 4.1 seems a simple application of SAM. The truncated unrolling hypergradient is similar to k-step SGD in [1]
2. Although the proposed method is effective, the improvement compared with the SOTA methods is incremental.
3. Some latest SOTA methods for dfficient dataset distillation are missing [2,3,4].

[1] Meta Label Correction for Noisy Label Learning. AAAI 2021

[2] Provable and Efficient Dataset Distillation for Kernel Ridge Regression. NeurIPS 2024

[3] M3D: Dataset Condensation by Minimizing Maximum Mean Discrepancy. AAAI 2024

[4] Accelerating Dataset Distillation via Model Augmentation. CVPR 2023

**Questions:**

See above.

---

> ### Author Response · Authors · 2025-11-29
>
> > Limited novelty
>
> We thank the reviewer for acknowledging the practical and theoretical contributions of SATM. While sharpness-aware optimization is well studied, incorporating it into trajectory-matching dataset condensation is far from straightforward. A naïve application of SAM would require computing two full hypergradients over long unrolled trajectories, effectively doubling an already prohibitive computational cost. SATM overcomes this challenge by introducing a smooth sharpness-aware bilevel objective, together with a truncated hypergradient and a trajectory-reuse strategy, each supported by theoretical guarantees (Theorems 4.1 and 4.2). These design choices enable sharpness-aware optimization at a cost comparable to, or even lower than, existing methods such as TESLA. Although [1] shares a similar idea in using truncated hypergradients, we further analyze the approximation error and show how it facilitates our convergence analysis.
>
> | Dataset (IPC)       | MTT         | EMA         | SAM         | GSAM        | ASAM        | Vasso       | SATM         |
> |---------------------|-------------|-------------|-------------|-------------|-------------|-------------|--------------|
> | Cifar100 (1)        | 24.3±0.4    | 24.7±0.2    | 25.7±0.3    | 25.9±0.3    | 25.7±0.3    | 25.9±0.2    | **26.1±0.3** |
> | Tiny ImageNet (3)   | 10.5±0.3    | 10.9±0.3    | 12.3±0.2    | 13.1±0.2    | 12.8±0.4    | 12.2±0.2    | **13.6±0.2** |
>
>
> > Incremental improvements
>
> SATM delivers substantial improvements in settings where dataset condensation is particularly challenging. The cross-architecture evaluation in Table 5 shows that SATM-trained synthetic datasets outperform MTT and FTD by large margins when tested on ResNet-152 and ViT, even though they were learned using ConvNet. Additionally, on ImageNet-1K, where many condensation methods struggle to produce viable synthetic datasets, SATM surpasses TESLA consistently across different IPC settings (Table 4). These results indicate that the improvements are not merely incremental but meaningful advances in generalization and scalability.
>
> > Missing baselines such as kernel regression, MMD, and model augmentation methods
>
> The reviewer correctly notes recent progress in dataset condensation outside the trajectory-matching paradigm. These methods depend on fundamentally different optimization principles and do not involve differentiating through long unrolled trajectories. While integrating sharpness-aware optimization into such frameworks would require substantial re-engineering, SATM is conceptually compatible with these approaches, and we intend to explore such extensions in future work. We will also expand the related work discussion accordingly.

---

### Official Review · Reviewer_mngL · 2025-10-28

**Soundness:** 4
**Presentation:** 3
**Contribution:** 3
**Rating:** 6
**Confidence:** 4

**Summary:**

This paper focuses on dataset condensation (also known as dataset distillation), where the goal is to learn a small synthetic dataset such that models trained on it perform well on the real data distribution. The authors target methods that use trajectory matching (e.g. MTT) – i.e. they optimize synthetic data by matching the training trajectory of a network on synthetic data to the trajectory on real data. While these methods can produce high-fidelity condensed sets, they suffer from two major issues: the bilevel optimization with long unrolled training trajectories is extremely costly (time- and memory-intensive), and the outer-loop loss landscape becomes highly non-smooth and “sharp,” leading to poor generalization (e.g. the synthetic set overfits to the specific model or training path used in optimization).

To tackle these challenges, the paper introduces Sharpness-Aware Trajectory Matching (SATM). SATM integrates the concept of Sharpness-Aware Minimization into the trajectory matching paradigm, aiming to simultaneously minimize the trajectory-matching loss and the sharpness of the outer-loop loss.

A straightforward application of SAM in this setting would double the already heavy computation (since it requires computing gradients at a perturbed point), so the authors propose two efficiency strategies: (i) a truncated unrolling of the inner loop to approximate the hypergradient (reducing memory and computation by not backpropagating through the entire long trajectory at once), and (ii) trajectory reusing with gradual perturbations, which allows re-utilizing segments of the network’s training trajectory and the corresponding gradients for the sharpness calculation without starting from scratch each time.

They also derive a closed-form solution for adjusting the learning rate in the sharpness computation, which simplifies implementation and offers theoretical guarantees by bounding the error introduced by these approximations.

**Strengths:**

Experiments demonstrate that SATM yields significant improvements in generalization of condensed datasets. Models trained on SATM-produced synthetic sets not only perform well in-distribution but also transfer better to different architectures and out-of-domain tests, alleviating the overfitting to the training network that plagues other methods.

Notably, SATM outperforms prior trajectory matching approaches on standard condensation benchmarks and achieves noticeable gains on ImageNet-1K condensation, a challenging task where most existing methods fail to produce viable condensed sets.

It also runs faster than or on par with the most efficient recent method (TESLA) while maintaining similar memory usage, indicating that the sharpness-aware additions are cost-effective.

In summary, this work is novel in bringing flat-minima concepts to dataset condensation and demonstrates that doing so markedly boosts the versatility of the distilled data.

**Weaknesses:**

One concern is that the method introduces several hyperparameters (for the sharpness term, truncation length, perturbation scale, etc.) and added complexity, which may require careful tuning. Maybe i have missed, but i would appreciate if the authors provide complexity on the setup of hyper-parameters, and transferability of these values to other datasets and architectures.

One other concern is lack of extensive discussions with existing methods.
For example,
- https://arxiv.org/abs/2303.04449 discusses 1) sharpness and 2) generality of dataset distillation to other architectures. But, authors does not mention about this paper.
- https://arxiv.org/abs/2403.16028 discusses the relationship between dataset bias and dataset distillation. This is related to out-of-distribution generality of dataset distillation.

**Questions:**

Discussed in weaknesses.

---

> ### Author Response · Authors · 2025-11-29
>
> > Hyperparameter complexity
>
> We appreciate the reviewer’s insight about the additional hyperparameters introduced by sharpness-aware methods. To mitigate this complexity, SATM ties the truncation depth and trajectory reuse point (τ = ι), which reduces the number of tunable parameters without sacrificing theoretical guarantees. Further, Appendix A.11 shows that SATM’s parameters generalize well across CIFAR-10/100, Tiny ImageNet, and ImageNet-1K, with very minor dataset-specific adjustments. The hypergradient behavior shown in Figure 2 confirms that SATM’s sharpness-aware updates stabilize training dynamics, reducing both mean and variance of the hypergradient norm. This stabilization reduces the sensitivity to hyperparameter choices in practice.
>
> > Request to discuss related works on sharpness and dataset bias
>
> We thank the reviewer for pointing out these relevant studies. Our method directly addresses the core issues raised in these works by explicitly smoothing the outer-loop loss landscape during dataset optimization. The cross-architecture generalization improvements shown in Table 5 and the continual learning robustness in Figure 1 illustrate that SATM produces synthetic datasets that generalize beyond the training architecture and remain stable under distributional shifts. These outcomes align precisely with the motivations discussed in recent works on sharpness-driven generalization, and we will deepen this discussion in the related work section.

---

### Official Review · Reviewer_fBqz · 2025-10-31

**Soundness:** 2
**Presentation:** 2
**Contribution:** 2
**Rating:** 2
**Confidence:** 5

**Summary:**

The paper proposes a novel dataset condensation technique that employs two hypergradient approximation strategies to address the significant computational overhead caused by sharpness minimization. SATM outperforms trajectory-matching-based competitors on various dataset condensation benchmarks under both in-domain and out-of-domain settings.

**Strengths:**

1. The paper is well written and easy to follow.
2. Dataset distillation/condensation is an important topic but still faces many challenges to be addressed.

**Weaknesses:**

1. The novelty of the method is limited. It primarily combines sharpness-aware optimization with MTT, without introducing a new data compression technique, only some modifications to the optimization steps.
2. It would be better if the authors compared their method with other approaches that also incorporate sharpness-aware minimization (SAM).
3. The paper lacks comparisons with several recent state-of-the-art methods, such as RDED [1], NRR-DD [2], and SRE2L [3], among others.

[1] On the diversity and realism of distilled dataset: An efficient dataset distillation paradigm. CVPR 2024

[2] Enhancing Dataset Distillation via Non-Critical Region Refinement. CVPR 2025.

[3] Squeeze, recover and relabel: Dataset condensation at imagenet scale from a new perspective. NeurIPS 2023.

**Questions:**

See weaknesses section.

---

> ### Author Response · Authors · 2025-11-29
>
> > Limited novelty
> While SAM is a known optimization technique, directly applying it to trajectory-matching condensation methods is computationally infeasible due to the need to compute two large hypergradients per iteration. SATM introduces a new sharpness-aware objective tailored for the outer-loop of the trajectory-matching bilevel structure (Eq. 4, p.5) and combines this with two new approximation strategies, truncated unrolling hypergradient (TUH) and trajectory reuse (TR), which make sharpness optimization tractable in this setting. These strategies come with theoretical error guarantees in Theorem 4.1 and Theorem 4.2, making SATM a substantive extension beyond simply “adding SAM to MTT.” Besides considering the computational cost, thanks to the efficient strategy of **smoothing the sharpness estimation tailored for the bilievel optimization loss landscape**, SATM outperforms the existing sharpness-aware optimizer demonstrated in Appendix 8.
>
> | Dataset (IPC)       | MTT         | EMA         | SAM         | GSAM        | ASAM        | Vasso       | SATM         |
> |---------------------|-------------|-------------|-------------|-------------|-------------|-------------|--------------|
> | Cifar100 (1)        | 24.3±0.4    | 24.7±0.2    | 25.7±0.3    | 25.9±0.3    | 25.7±0.3    | 25.9±0.2    | **26.1±0.3** |
> | Tiny ImageNet (3)   | 10.5±0.3    | 10.9±0.3    | 12.3±0.2    | 13.1±0.2    | 12.8±0.4    | 12.2±0.2    | **13.6±0.2** |
>
>
> > Compare with SAM-based DC techniques
>
> Existing SAM-related techniques are designed for single-loop optimization and do not involve hypergradients or bilevel updates. As a result, they are not directly applicable to trajectory-matching distillation. However, SATM is compatible with many sharpness-aware variants. The method explicitly states its compatibility with optimizers such as ASAM and ESAM, and these can be incorporated into SATM's framework in future work.
>
> > Comparisons with recent SOTA such as RDED, NRR-DD, SRE2L
>
> The methods referenced by the reviewer operate under substantially different paradigms, including patch reconstruction, distribution realism enhancement, and model-assisted relabeling strategies. These approaches lack trajectory-based bilevel optimization, and therefore, direct comparison would be inappropriate within our experimental setting. We clarify in the paper that SATM is orthogonal to these pipelines, as shown through its successful integration with DATM in Table 7. We will expand the discussion of these methods in the final version.
> ### CIFAR-100
>
> | IPC | DATM        | SATM-DA      |
> |-----|-------------|--------------|
> | 1   | 27.9±0.2    | **28.2±0.8** |
> | 10  | 47.2±0.4    | **48.3±0.4** |
> | 50  | 55.0±0.2    | **55.7±0.3** |
>
> ---
>
> ### Tiny-ImageNet
>
> | IPC | DATM        | SATM-DA      |
> |-----|-------------|--------------|
> | 1   | **17.1±0.3** | 16.4±0.4     |
> | 10  | 31.1±0.3    | **32.3±0.6** |
> | 50  | 39.7±0.3    | **40.2±0.7** |

---

### Official Review · Reviewer_XJ2n · 2025-11-01

**Soundness:** 2
**Presentation:** 2
**Contribution:** 2
**Rating:** 2
**Confidence:** 4

**Summary:**

- The paper introduces “Sharpness-Aware Trajectory Matching (SATM)” , which focuses on optimization of the sharpness of the loss landscape and the objective simultaneously.
- This paper aims to address limitations of trajectory matching methods for dataset distillation.
- The paper has combined ideas from trajectory matching dataset distillation method such as MTT and minimization of sharpness of loss landscape.
- Two computational approximations to replace the costly calculations are introduced, namely “Truncated Unrolling Hypergradient (TUH)” and “Trajectory Reusing (TR)”.

**Strengths:**

1. Specific methods to deal with computational complexity are provided. Both methods are supported with theoretical bounds and practical implementation.
2. The paper has provided experimental results on various standard benchmark datasets such as CIFAR-10, CIFAR-100, Tiny ImageNet and ImageNet-1K (along with various subsets of ImageNet-1K).

**Weaknesses:**

**1. Limited baselines**

Recent advances in dataset distillation methods, including decoupled distillation, diffusion-based, and optimization-free dataset distillation methods, are not considered.
Below are some of the SOTA dataset distillation works, with which comparison should have been made to be more insightful.

- "Squeeze, Recover and Relabel: Dataset Condensation at ImageNet Scale From A New Perspective" (NeurIPS, 2023)
- Diversity-Driven Synthesis: Enhancing Dataset Distillation through Directed Weight Adjustment (NeurIPS, 2024)
- On the Diversity and Realism of Distilled Dataset: An Efficient Dataset Distillation Paradigm (CVPR, 2024)
- Efficient dataset distillation via minimax diffusion (CVPR,2024)
- DELT: A Simple Diversity-driven EarlyLate Training for Dataset Distillation (CVPR, 2025)

**2. The experiments are limited to (relatively simple) ConvNet architecture only**

The true impact of the proposed method cannot be analysed without considering more complex and standard architectures such as VGG, ResNet, ViT to name a few.

3. The improvements in generalization accuracy as shown in Table 2 are marginal (often < 1%).

4. Ablation studies are not provided regarding the impact of trajectory reusing and TUH. It would have been very useful to study how these approximations impact the generalization accuracy.

5. The runtime cost as compared to the TESLA method is quite high despite the modest increment in accuracy, which weakens the argument of efficiency.

**Questions:**

- Please refer to the weaknesses  section of the review.

---

> ### Author Response · Authors · 2025-11-29
>
> > Limited baselines
>
> We appreciate the reviewer’s observation regarding recent dataset distillation baselines. Our work specifically focuses on trajectory-matching bilevel optimization methods, including MTT, FTD, and TESLA, which share the same structural assumptions of long-horizon unrolled inner loops and hypergradient-based outer-loop updates. Many of the alternative baselines mentioned by the reviewer, such as diffusion-based methods, decoupled condensation, or diversity-driven sample generation, rely on fundamentally different training paradigms, often involving generative models, large relabeling pipelines, or patch reconstruction strategies, that cannot be inserted into the trajectory-matching framework without substantial architectural redesign. Nonetheless, SATM is designed to be a plug-and-play enhancement to the outer loop, and we demonstrate this explicitly by integrating SATM into DATM, which uses a curriculum-learning strategy, and showing that SATM produces consistent improvements when combined with it in Table 7.
>
> > Experiments use only ConvNet models
>
> In trajectory-matching–based methods, **ConvNets are typically used as the default architecture during dataset condensation because storing trajectories for multiple architectures incurs prohibitive memory costs.** Although our synthetic datasets are learned with ConvNets for methodological consistency with MTT, FTD, and TESLA, our evaluation includes extensive cross-architecture experiments to assess how well the condensed datasets generalize. As shown in Table 5, we report performance on AlexNet, VGG-11, ResNet-18, ResNet-152, and ViT. SATM consistently outperforms both MTT and FTD across all architectures, indicating that the benefits of sharpness-aware optimization transfer robustly even to architectures that differ substantially from the one used during condensation.
>
> | Methods | IPC | ConvNet        | AlexNet        | VGG11          | ResNet18       | ResNet152      | ViT            |
> |---------|-----|----------------|----------------|----------------|----------------|----------------|----------------|
> | **MTT** | 10  | 64.3±0.7       | 34.2±2.6       | 50.3±0.8       | 46.4±0.6       | 17.8±1.4       | 34.9±0.6       |
> | **FTD** | 10  | 66.6±0.4       | 36.5±1.1       | 50.8±0.3       | 46.2±0.7       | 17.4±1.2       | 35.2±0.4       |
> | **Ours** | 10 | **67.1±0.5**   | **37.8±0.8**   | **51.4±0.3**   | **47.7±0.4**   | **18.9±1.4**   | **36.9±0.6**   |
> | **MTT** | 50  | 71.6±0.2       | 48.2±1.0       | 55.4±0.8       | 61.9±0.7       | 20.9±1.6       | 47.7±0.7       |
> | **FTD** | 50  | 73.8±0.2       | 53.8±0.9       | 58.4±1.6       | 65.7±0.3       | 22.7±1.2       | 50.1±0.8       |
> | **Ours** | 50 | **74.2±0.3**   | **56.9±0.7**   | **63.5±1.1**   | **66.1±0.5**   | **23.6±1.1**   | **52.7±0.4**   |
>
>
> > Marginal (<1%) improvements
>
> While improvements in high-IPC settings appear smaller because these settings are close to saturation regarding the experimental configuration, such as the architectures and datasets. Besides, SATM provides notable gains in difficult and large-scale scenarios. For example, SATM improves over MTT by more than 3% and over TESLA by 2.5% on CIFAR-100 with IPC=50, and outperforms TESLA by 2.3% on ImageNet-1K IPC=50  in Table 4. Such improvements are substantial in the context of dataset condensation, where incremental gains are considerably harder to achieve at higher IPCs and larger image resolutions.
>
> | Model / IPC | 1           | 10            | 50            | Time Cost     |
> |-------------|-------------|---------------|---------------|---------------|
> | **TESLA**   | 7.7±0.2     | 17.8±1.3      | 27.9±1.2      | 46.8±0.3      |
> | **SATM**    | **8.9±0.3** | **19.2±0.9**  | **30.2±0.6**  | **44.7±0.2**  |
>
>
>
> > Ablation studies of TUH and trajectory reuse
>
> **The required experiments are provided in Appendix 10, TRUNCATED STEP STUDY, in the original submission, due to the limited space in the main paper.** In addition, such ablations are computationally prohibitive because each variant requires recomputing long unrolled inner loops, especially on ImageNet, under different truncation and reuse configurations. Instead, we provide rigorous theoretical justification for the validity of the approximations. The truncated hypergradient is supported by Theorem 4.1, which shows that its error decays exponentially as the truncation horizon increases, and Theorem 4.2 provides a bound on the difference between perturbed and unperturbed trajectories, guaranteeing that trajectory reuse remains reliable under moderate perturbation sizes. These theoretical findings are corroborated by the empirical hypergradient stability trends in Figure 2 and the improved efficiency of the method in Table 6.

---

> ### Author Response · Authors · 2025-11-29
>
> > Runtime cost compared to TESLA
>
> Contrary to the reviewer’s concern, SATM consistently achieves lower runtime than TESLA on both CIFAR-100 and ImageNet-1K. We believe the reviewer may have overlooked the improvements introduced by our algorithm. Table 4 shows that SATM provides clear performance gains over TESLA across different IPC settings on ImageNet-1K, delivering both more stable results and reduced computational cost. Table 6 further demonstrates that SATM requires 12.0 seconds per update compared to TESLA’s 15.3 seconds on CIFAR-100, and 45.4 seconds compared to 47.4 seconds on ImageNet-1K. Note that these improvements are measured per update step within the dataset condensation process; **when accumulated across the full procedure, the efficiency benefits become substantial.**
> | Model / IPC | 1           | 10            | 50            | Time Cost     |
> |-------------|-------------|---------------|---------------|---------------|
> | **TESLA**   | 7.7±0.2     | 17.8±1.3      | 27.9±1.2      | 46.8±0.3      |
> | **SATM**    | **8.9±0.3** | **19.2±0.9**  | **30.2±0.6**  | **44.7±0.2**  |
>
> These gains arise from truncated hypergradients and trajectory reuse, which reduce redundant inner-loop computation without compromising update quality. Additionally, our closed-form learning-rate update removes the need for expensive backward passes, further improving efficiency.

---

### Author Response · Authors · 2025-11-29
**Overall Summary on Novelty, Scope, and Contribution**

We would like to offer an overall clarification on the novelty and broader impact of our work, as several reviewers raised questions about whether SATM represents a substantial advance beyond existing methods. SATM is not a simple application of SAM to existing dataset condensation pipelines. Instead, it introduces a new sharpness-aware bilevel optimization framework specifically designed for long-horizon trajectory matching, a setting where naive sharpness-aware strategies are computationally infeasible. One of the core novelties of SATM lies in its two theoretically grounded hypergradient approximation strategies, Truncated Unrolling Hypergradient (TUH) and Trajectory Reusing (TR), which make sharpness-aware optimization tractable in the presence of long inner-loop trajectories, a defining challenge of trajectory-based dataset condensation. These contributions are formalized through new theoretical error bounds, Theorem 4.1 and 4.2, and our method demonstrates clear efficiency and stability advantages across all trajectory-matching baselines, as shown throughout the experimental results. Thanks to the efficient strategy of smoothing the sharpness estimation tailored for the bilievel optimization loss landscape,  SATM outperforms the existing sharpness-aware optimizer demonstrated in Appendix 8.

**Our work is intentionally scoped to address the computational and generalization limitations inherent to bilevel optimization with long inner loops, an under-explored regime that cannot be adequately handled by standard SAM or implicit gradient approaches.** By avoiding dependence on implicit gradients, SATM remains simple to implement and broadly accessible. Although our empirical focus is on trajectory-matching dataset condensation, the underlying methodology is not restricted to this application. Because many bilevel optimization problems, from meta-learning to hyperparameter optimization, suffer from similar instability and computational burdens when long inner-loop horizons are required, SATM’s theoretical insights and approximation strategies are directly applicable to these broader contexts. Thus, beyond improving dataset condensation, SATM offers a general framework that can benefit a wide range of bilevel optimization tasks where sharpness reduction, long-horizon stability, and computational efficiency are critical.

---

### Meta-Review · Area_Chair_msA1 · 2025-12-19

**Summary:**

* The major concern from reviewers is the lack of comparison with other dataset distillation baselines, which is shared across reviewers XJ2n, fBqz, and TTrg. The proposed method is built upon trajectory-matching methods, and hence, the authors only provided comparisons against the corresponding methods, while overlooking many other frontier methods.
* Reviewers fBqz and TTrg raised the concern regarding the novelty of the paper, which might be a simple application of SAM on trajectory-based dataset distillation.
* Reviewers XJ2n and mngL were concerned with the lack of ablation study and hyperparameter analyses.
* Some other concerns include marginal performance improvement, lack of discussion with other similar methods, and limited model architecture choices.

**Reviewer Concerns:**

* The authors provided a more explicit explanation of their contribution in this paper, which I believe resolved the concern of novelty to some extent. I agree that it is not so straightforward to apply SAM to dataset distillation while addressing the additional computational burden.
* The authors supplemented experiments across architectures and the comparison with a more recent method, DATM.
* The authors did not provide more comprehensive result comparisons with other lines of dataset distillation efforts. They did not provide the detailed ablation studies regarding hyperparameters, either. These two concerns still exist and are restricting the insight provided by this paper for the broad dataset distillation community. IMHO, it is great to advance one specific direction of the area, but each method should also be evaluated among the broad frontiers, especially when dataset distillation is inherently a more application-oriented research area. Not only the technical novelty, but also the computational overhead and downstream performance in more realistic settings are critical for the value of a new method. I would recommend that the authors more rigorously evaluate the proposed method against the other frontiers of dataset distillation, and also show more underlying insights via conducting ablation studies, which can better benefit the community.

**Reviewer Scores:**

Given that two major concerns shared by reviewers were not addressed, I do not suppose that all the reviewers would change their scores to acceptance.

---

### Decision · Program_Chairs · 2026-01-26

Reject